# Synonymous Codon Usage Bias in the Chloroplast Genomes of 13 Oil-Tea Camellia Samples from South China

**Jing Chen, Wuqiang Ma, Xinwen Hu and Kaibing Zhou ***

Sanya Nanfan Research Institute, Hainan University, Sanya 572025, China; kimchen111@163.com (J.C.); wuqiangma19865@foxmail.com (W.M.); huxinwen@hainanu.edu.cn (X.H.)
\* Correspondence: zkb@hainanu.edu.cn

**Abstract:** Synonymous codon usage (SCU) bias in oil-tea camellia cpDNAs was determined by examining 13 South Chinese oil-tea camellia samples and performing bioinformatics analysis using GenBank sequence information, revealing conserved bias among the samples. GC content at the third position (GC3) was the lowest, with a preference for A or T, suggesting weak SCU bias. The GC contents at the first two codon positions (GC1 and GC2) were extremely significantly correlated with one another but not with the expected number of codons (ENC). GC3 was not correlated with GC1 and GC2 but was extremely significantly correlated with ENC. Of the 30 high-frequency codons, 15, 14, 1 and 0 codons had U, A, G and C at the third position, respectively. The points for most genes were distributed above the neutrality plot diagonal. The points for 20 genes, accounting for 37.74% of all coding sequences (CDSs), were distributed on or near the ENC plot standard curve, and the ENC ratio ranged from −0.05–0.05. However, those of the other genes were under the standard curve, with higher ENC ratios. The points for most genes were distributed in the lower part of the PR2 plot, especially the bottom right corner. Twenty-eight highly expressed codons were screened and 11, 9, 7 and 1 codons had U, A, C and G as the third base, respectively. Twenty optimal codons were screened by comparing high-frequency codons and 11, 8, 0 and 1 codons had U, A, C and G as the third base, respectively. All samples were divided into six clades ($r^2$ = 0.9190, d = 0.5395) according to a relative synonymous codon usage (RSCU)-based phylogenetic tree. *Camellia gauchowensis*, *C. vietnamensis*, an undetermined oil-tea camellia species from Hainan province, and *C. osmantha* belonged to the same clade; the genetic relationships between *C. gauchowensis*, *C. vietnamensis* and the undetermined species were the closest. In summary, SCU bias is influenced by selection, while the influence of mutation cannot be ignored. As the SCU bias differed between species, this feature can be used to identify plant species and infer their genetic relationships. For example, *C. vietnamensis* and *C. gauchowensis* can be merged into one species, and the undetermined species can be considered *C. vietnamensis*. The results described here provide a basis for studying cpDNA gene expression and the development of cpDNA genetic engineering.

**Keywords:** *Camellia*; cpDNA; RSCU; mutation; selection; optimal codon

## 1. Introduction

There are 20 amino acids, yet there are 61 codons that encode them [1] and it is known that synonymous codons encode 18 different amino acids, but the usage of these synonymous codons is biased; meanwhile, the optimal codons can be identified from synonymous codon usage (SCU) bias [2]. SCU is thought to be an evolutionary behavior for organisms to adapt to their environment [3], and it has been reported that mutation, selection and drift are the main reasons for SCU [4]; this is the most important issue to be debated. SCU research has been helpful in elucidating the molecular evolution and adaptation to the environment and thus the evolutionary relationships among different species [5]. Additionally, SCU research has been helpful in gene expression research [6] and genetic engineering [7].

Chloroplast genomes (cpDNAs) are generally covalently closed circular DNA with a genome size of 115~165 kb, which exists as multiple copies in cells and can be transcribed and translated during gene expression [8]. Due to their advantages of stable structure, conserved gene content and slow molecular evolution rate, cpDNAs have been widely used in research on species evolution, species classification and phylogeny [9].

In summary, analyzing the SCU of cpDNAs seems to be one of the best methods for researching the evolutionary relationships among different species of plants. To date, research on the SCU of cpDNAs has been reported in *Phalaenopsis aphrodite* [10], *Gossypium hirsutum* [11], *Camellia oleifera* [12], *Gelidocalamus tessellatus* [13] and *Trollius chinensis* [14], among others.

Oil-tea camellia plants belong to the *Camellia* genus and produce seeds containing large quantities of oil. These economically important plants are cultivated in many areas and distributed in 18 provinces or autonomous regions of South China. The oil produced by these plants is an important and unique high-value food oil [15]. High-yield cultivars for afforestation have not yet been selected in the oil-tea camellia plants in South China, such as *C. vietnamensis*, *C. gigantocarpa* and *C. osmantha*, in contrast to *C. oleifera* and *C. meiocarpa*; it is therefore extremely urgent to study issues related to the development of oil-tea camellia germplasm and the breeding of oil-tea camellia [16].

Studying the SCU bias in cpDNAs of different oil-tea camellia germplasms provides a new framework for understanding the genetic evolution of *Camellia* plants [12,17]. Researching the causes of SCU bias helps predict the expression efficiency of the cpDNA genes of oil-tea camellia plants, guides the development of cpDNA genetic engineering and aids in the construction of a technological system for the molecular breeding of oil-tea camellia plants [18]. Therefore, it is essential to study SCU bias in the cpDNA of oil-tea camellia plants.

Our research group sequenced the cpDNAs of 13 oil-tea camellia plant samples and analyzed and compared the structures of all cpDNAs. Then, the specificity of the cpDNAs of the different species was examined, and the identification of undetermined species of oil-tea camellia plants from Hainan province was carried out. All samples belonged to six species of oil-tea camellia, including *C. vietnamensis*, whose samples were collected from production areas in seven different counties or cities [19]. Further research on the SCU bias in the cpDNAs of 13 oil-tea camellia samples was performed to reveal the genetic relationships and cpDNA gene expression of these plants.

## 2. Materials and Methods

### 2.1. Experimental Materials

Leaf samples for cpDNA sequencing were collected from various plants, including 13 samples whose information is shown in Table 1. According to assembly and comparison after cpDNA sequencing, the cpDNA sequences of the HD10~HD13 samples (3 undetermined species from Hainan province and *C. gauchowensis* from Xuwen county, Guangdong Province) were identical, and four emerged from the cpDNA of HD10 [19]. Therefore, 10 cpDNAs were used to analyze SCU bias. Based on the annotation of the 10 cpDNAs and referring to the genome of *C. oleifera* (HD07, MN078090), 53 efficient coding sequences (CDSs) were screened to analyze SCU bias after deleting those with lengths less than 300 bp, repeat genes and termination codons.

### 2.2. Calculations of the GC Content and ENC Value

Using CUSP software, the contents of GC at the first, second and third positions of each codon of each gene were calculated (namely, GC1, GC2 and GC3, respectively), and the total GC content (GCall) and the effective number of codons (ENC) value of each codon of each gene were determined. ENC values range from 20 to 61, with 20 indicating extreme bias and only one codon used for each amino acid (AA) and 61 indicating no bias, with all synonymous codons used for each amino acid [20]. One-unit linear correlation analysis

was performed in the R environment with a two-tailed test, with the symbol ** indicating extreme significance at $p \leq 0.01$ and the symbol * indicating significance at $p \leq 0.05$.

**Table 1.** Basic information on the different oil-tea camellia species.

| Forestland | Plant's Site | Species | | Tree Age/a | Sample Symbol |
|---|---|---|---|---|---|
| | | **Common Name** | **Latin Name** | | |
| Wangsha village, Changpo town, Gaozhou city, Guangdong province | 22°0′40.87″ N 111°6′25.49″ E | Gaozhou population of Gaozhou oil-tea camellia | *Camellia gauchowensis* Chang | >40 | HD01 |
| Guanshan village, Shahu town, Luchuan county, Guangxi Zhuang Autonomous Region | 22°21′48.27″ N 110°12′20.55″ E | Luchuan population of Gaozhou oil-tea camellia | *Camellia gauchowensis* Chang | >40 | HD02 |
| Youbang village, Nalin town, Bobai city, Guangxi Zhuang Autonomous Region | 22°14′7.45″ N 109°43′53.85″ E | Bobai large fruit oil-tea camellia | *Camellia gigantocarpa* Hu et T. C. Huang | >40 | HD03 |
| Guangxi Research Institute of Forestry | 22°55′13.45″ N 108°21′3.85″ E | Wantian red flower oil-tea camellia | *Camellia polyodonta* How ex Hu | 13 | HD04 |
| Guangxi Research Institute of Forestry | 22°55′13.45″ N 108°21′3.85″ E | Small fruit oil-tea camellia | *Camellia meiocarpa* Hu | >40 | HD05 |
| Guangxi Research Institute of Forestry | 22°55′13.45″ N 108°21′3.85″ E | Guangning red flower oil-tea camellia | *Camellia semiserrata* Chi. | 16 | HD06 |
| Guangxi Research Institute of Forestry | 22°55′13.45″ N 108°21′3.85″ E | Common oil-tea camellia | *Camellia oleifera* Abel. | >40 | HD07 |
| Guangxi Research Institute of Forestry | 22°55′13.45″ N 108°21′3.85″ E | Xianghua oil-tea camellia | *Camellia osmantha* Ye CX, Ma JL et Ye H | 13 | HD08 |
| Guangxi Research Institute of Forestry | 22°55′13.45″ N 108°21′3.85″ E | Vietnam oil-tea camellia | *Camellia vietnamensis* T. C. Huang ex Hu | | HD09 |
| Zhongjiu village, Huishan town, Qionghai city, Hainan province | 19°5′18.30″ N 110°18′18.29″ E | Hainan oil-tea camellia | Undetermined species | >600 | HD10 |
| Xingwen village, Wangwu town, Danzhou city, Hainan province | 19°40′22.66″ N 109°20′48.84″ E | Hainan oil-tea camellia | Undetermined species | >195 | HD11 |
| Zaha village, Changhao region, Wuzhishan city, Hainan province | 18°40′31″ N 109°27′56″ E | Hainan oil-tea camellia | Undetermined species | >40 | HD12 |
| Andong village, Longtang town, Xuwen county, Guangdong province | 20°18′32.66″ N 110°20′44.86″ E | Xuwen population of Gaozhou oil-tea camellia | *Camellia gauchowensis* Chang | >40 | HD13 |

### 2.3. Analysis of Relative Synonymous Codon Usage (RSCU)

The formula for calculating RSCU values was as follows.

RSCU = Observed frequency of a codon/Expected frequency under the assumption that all synonymous codons for those amino acids are used equally.

RSCU was calculated with CodonW 1.4.2 software, and a corresponding plot was drawn with Microsoft Office Excel 2016 software. Cluster analysis was carried out with the single method by using the cluster procedure in SAS based on the RSCU of each synonymous codon.

### 2.4. Neutrality Plot Construction

A neutrality plot is used primarily to identify the factors influencing SCU bias [21]. The content of GC3 and the content of GC12 [GC12 = (GC1 + GC2)/2] are the horizontal and vertical ordinates, respectively, and a two-dimensional scatter diagram in which each point symbolizes a particular gene is drawn. If the points are distributed along the diagonal, the linear regression is near 1, then the GC12 and GC3 contents are essentially the same. In other words, the base compositions at the different positions in the codon are almost the same, indicating that the gene is only slightly influenced by selection pressure but strongly influenced by mutation pressure [22]. If the points are distributed far away from the diagonal, the linear regression approaches 0, meaning that the difference in the GC12 and GC3 contents is strong, i.e., that the gene is influenced mainly by selection pressure [23].

### 2.5. ENC Plot Construction

The ENC is used to identify the range of SCU bias. The expected value ranges from 20 to 61, where values closer to 20 indicate that SCU bias is influenced more by mutation pressure and, otherwise, more by selection pressure [24]. The content of GC in the third position of the synonymous codons (GC3s) in each cpDNA CDS and the actual value

of ENC (ENCcobs) are the horizontal and vertical ordinates, respectively, and a two-dimensional scatter diagram is drawn. The curve of the ENC expected value (ENCexp) is drawn according to the formula ENCexp = 2 + GC3s + 29/[GC3s2 + (1 − GC3s)2]. Then, the ENCexp of each CDS is calculated based on the content of GC3s of each CDS [20], and the ratio of ENC (ENCratio) is calculated according to the formula ENCratio = (ENCexp-ENCcobs)/ENCexp [25].

### 2.6. PR2 Plot Construction

The analysis of PR2 plots is also called analysis of parity preference; it reveals whether the difference in the combination of the 4 bases, i.e., A, T, C and G, at the third position of a codon influences SCU bias [26]. In this study, the four degenerate codons with variation only at the third position for the amino acids (valine, proline, threonine, alanine, glycine, serine, leucine and arginine) were also used for PR2 evaluation [27]. The ratio of the G3s content to the sum of the G3s and C3s contents is the horizontal ordinate, and the ratio of the A3s content to the sum of the A3s and T3s contents is the vertical ordinate. Then, a two-dimensional scatterplot is drawn and analyzed. Points in the center show that the base content is even, i.e., that A = T and C = G, indicating that there is no parity preference or mutation, while the vector from the center point indicates the degree and direction of the base shift [28].

### 2.7. Optimal Codon Analysis

The high-frequency codons whose RSCU was more than 1 were chosen [29], while the 53 CDSs were arranged from high to low in terms of ENC value. Then, 10% of the genes were chosen from the highest and lowest ends, and the high- and low-expression gene groups were identified. For each codon, the RSCU of the high-expression group minus that of the low-expression group was calculated, and the difference was symbolized with ΔRSCU. If the codon's ΔRSCU value was not less than 0.08, it was regarded as a high-expression codon [4]; eventually, the optimal codons were determined by comparing the high-frequency codons and the high-expression codons.

### 2.8. Construction of a cpDNA Phylogenetic Map

A total of 65 complete cpDNA sequences were obtained according to the method of a previous report [19], and 10 sample sequences were added for analysis. CDSs of the above 75 full cpDNA sequences were extracted for phylogenetic analysis. The phylogenetic tree was constructed according to the method in that same report [19]. The optimal model was selected through the IQ-TREE model finder, and the optimal mode was GTR + invgamma. The phylogenetic tree was constructed using IQ-TREE version 2 software. The outgroup was set as Hartia_laotica (NC_041509.1), and the IQ-TREE parameters were set as -BB 1000 and -ALRT 1000, which denoted a nucleic acid molecule replacement model set as GTR. Rate variation across sites was defined by the invgamma model; the prior probability model parameters were set to default values; and the parameters for Markov chain Monte Carlo sampling were Nruns = 2, Nchain = 4, Ngen = 1,000,000, Samplefreq = 500 and Temp = 0.05, indicating that the CDSs in the analysis were run simultaneously. When the original tree results were obtained, the branches unrelated to the sample were removed to obtain the final phylogenetic tree.

## 3. Results and Analysis

### 3.1. Base Composition of Oil-Tea Camellia cpDNAs

The same 53 cpDNA CDSs were screened from each sample, and the base compositions of these CDSs are shown in Table 2.

**Table 2.** GC content at different positions and ENC values of codons in the chloroplast genome of oil-tea camellia.

| Gene Category | Gene Group | Gene | GC1 | GC2 | GC3 | GCall | ENC | Plants |
|---|---|---|---|---|---|---|---|---|
| Genes for photosynthesis | ATP synthase | *atpA* | 55.51 | 40.16 | 23.82 | 39.83 | 42.89 | |
| | | *atpB* | 56.71 | 41.48 | 28.06–28.26 | 42.08–42.15 | 44.79–44.99 | HD04, HD05, HD07 (28.06), others (28.26) |
| | | *atpE* | 50.75 | 38.06 | 27.61 | 38.81 | 47.78 | |
| | | *atpF* | 45.95 | 34.05 | 35.68–36.22 | | | HD03 (36.22), others (35.68) |
| | | *atpI* | 49.19 | 37.90 | 26.61 | 37.90 | 44.59 | |
| | Cytochrome b/f complex | *petA* | 52.34 | 37.07 | 28.04 | 39.15 | 48.56 | |
| | | *petB* | 48.61 | 41.67 | 30.56 | 40.28 | 42.69 | |
| | | *petD* | 50.93 | 39.13 | 26.09 | 38.72 | 43.64 | |
| | NADH dehydrogenase | *ndhA* | 42.03 | 39.01 | 20.33–20.60 | 33.79–33.88 | 41.10–41.23 | HD01, HD09 (20.33), others (20.60) |
| | | *ndhB* | 41.68 | 38.36 | 30.92–31.12 | 36.99–37.05 | 46.46–46.73 | HD01, HD02, HD08, HD09, HD10 (30.92), others (31.12) |
| | | *ndhC* | 46.28 | 33.88 | 24.79 | 34.99 | 45.99 | |
| | | *ndhD* | 40.12 | 37.18 | 26.61–26.42 | 34.57–34.64 | 45.88–46.31 | HD08, others (26.61) |
| | | *ndhE* | 39.22 | 32.35–33.33 | 24.51 | 32.03–32.35 | 40.93–41.09 | HD07 (32.35), others (33.33); |
| | | *ndhF* | 36.58–36.85 | 35.65–35.78 | 22.70–22.83 | 31.69–31.78 | 41.66–41.73 | HD04, HD05 (36.58), 08 (36.85), others (36.72); HD03~HD07 (35.78), others (35.65); HD03~HD05 (22.83), others (22.70) |
| | | *ndhG* | 41.81–42.37 | 32.77 | 22.03 | 32.20–32.39 | 42.76–42.88 | HD03, HD04, HD05, HD07 (42.37), others (41.81) |
| | | *ndhH* | 50.76 | 36.04–36.29 | 24.37–24.62 | 37.06–37.23 | 46.64–46.69 | HD03, HD04, HD05, HD07 (36.04, 24.37), others (36.29, 24.62) |
| | | *ndhI* | 42.26 | 36.29–37.50 | 27.98 | 35.71–35.91 | 48.87–49.81 | HD01, HD08, HD10 (36.29), others (27.50) |
| | | *ndhJ* | 50.31 | 37.74 | 30.82–31.45 | 39.62–39.83 | 50.00–51.41 | HD03 (30.82), others (31.45) |
| | | *ndhK* | 42.98–44.84 | 41.32–43.05 | 22.87–23.55 | 35.95–36.92 | 47.22–47.24 | HD01 (42.98, 41.32, 23.55), HD03 (44.35, 42.61, 23.48), others (44.84, 43.05, 23.87) |
| | Photosystem I | *psaA* | 52.20 | 43.54 | 31.56–31.69 | 42.43–42.48 | 49.13 | HD01, HD02, HD08, HD09, HD10 (31.56), others (31.69) |
| | | *psaB* | 48.84 | 42.99 | 30.75 | 40.86 | 47.80 | |
| | Photosystem II | *psbA* | 49.72 | 43.50 | 32.20–32.49 | 41.90 | 40.60 | HD01, HD09 (32.20), others (32.49) |
| | | *psbB* | 55.01 | 46.17 | 30.84–31.24 | 44.01–44.07 | 47.14–47.29 | HD01, HD09, HD10 (30.84), HD03, HD04, HD05, HD06 (31.24), others (31.04) |
| | | *psbC* | 53.16 | 46.41 | 31.43 | 43.67 | 43.75 | |
| | | *psbD* | 51.69 | 43.22 | 31.36 | 42.09 | 43.19 | |
| | Rubisco large subunit | *rbcL* | 58.61 | 43.70 | 30.04 | 44.12 | 48.16 | |
| | ATP-dependent protease subunit p gene | *clpP* | 58.67 | 37.76 | 25.51 | 40.65 | 49.00 | |

**Table 2.** *Cont.*

| Gene Category | Gene Group | Gene | GC1 | GC2 | GC3 | GCall | ENC | Plants |
|---|---|---|---|---|---|---|---|---|
| Self-replication | Ribosomal proteins (LSU) | *rpl14* | 56.10 | 36.59 | 26.02 | 39.57 | 44.21 | |
| | | *rpl16* | 51.47 | 52.21–52.94 | 19.12 | 40.93–41.18 | 35.05–35.23 | HD06 (52.94), others (52.21) |
| | | *rpl2* | 50.18 | 47.64 | 32.36 | 43.39 | 54.12 | |
| | | *rpl20* | 38.98 | 43.22 | 25.42–26.27 | | | HD08 (26.27), others (25.42) |
| | | *rpl22* | 41.03 | 37.18 | 25.00 | 34.40 | 43.00 | |
| | RNA polymerase | *rpoA* | 44.64 | 32.14–32.44 | 24.70–25.00 | 33.83–34.03 | 48.56–48.80 | HD04 (32.44), others (32.14); HD02~HD05, HD07 (25.00), others (24.70) |
| | | *rpoB* | 50.14–50.33 | 38.00–38.75 | 27.73–27.82 | 38.66–38.75 | 48.31–48.37 | HD04 (50.23), HD07 (50.14), others (50.33); HD01 (38.75), HD08~HD10 (38.10), others (38.00); HD04, D05 (27.73), others (27.82) |
| | | *rpoC1* | 49.85–50.00 | 37.72 | 28.22–28.36 | 38.65–38.69 | 50.08–50.15 | HD03 (49.85), others (50.00);HD04 (28.22), others (28.36) |
| | | *rpoC2* | 45.60–45.81 | 37.82–37.87 | 28.44–28.58 | 37.28–37.41 | 49.18–49.29 | HD04, HD07 (45.75), HD05, HD06 (45.67), others (45.60);HD03 (37.87), others (37.82); HD01, HD09, HD10 (28.58), HD02, HD06 (28.51), HD03, HD05 (28.55), HD04, HD07, HD08 (28.44) |
| | Ribosomal proteins (SSU) | *rps11* | 52.52 | 57.55 | 20.86–21.58 | 43.65–43.88 | 47.80–48.58 | HD06 (21.58), others (20.86) |
| | | *rps12* | 52.10 | 50.42 | 29.41 | 43.98 | 50.23 | |
| | | *rps14* | 43.56 | 47.52 | 31.68 | 40.92 | 37.46 | |
| | | *rps18* | 35.29 | 43.14 | 26.47–25.49 | 34.64–34.97 | 34.68–35.64 | HD06 (25.49), others (26.47) |
| | | *rps2* | 43.46 | 42.19 | 28.27–27.85 | 37.83–37.97 | 47.62–47.85 | HD04, HD07 (27.85), others (28.27) |
| | | *rps3* | 47.03 | 31.51 | 22.83 | 33.79 | 47.33 | |
| | | *rps4* | 50.00 | 37.13 | 25.74 | 37.62 | 47.88 | |
| | | *rps7* | 51.92 | 45.51 | 23.08 | 40.17 | 45.81 | |
| | | *rps8* | 40.44–42.65 | 41.18 | 27.21 | 36.27–37.01 | 40.57–41.79 | HD01, HD06, HD08, HD09, HD10 (41.18), HD02 (40.44), HD03, HD05 (41.91), HD04, HD07 (42.65) |
| Other genes | Subunit of acetyl-CoA-carboxylase | *acc*D | 40.44–40.64 | 35.81–36.02 | 29.38–29.58 | 35.21–35.35 | 47.84–48.28 | HD01, HD09 (40.64), others (40.44);HD04~HD07 (36.02), others (35.81);HD01, HD08~HD10 (29.58), others (29.38) |
| | c-type cytochrome synthesis ccsA gene | *ccs*A | 33.54 | 36.96 | 24.22–24.84 | 31.57–31.78 | 47.01–47.46 | HD08 (24.84), others (24.53) |
| | Maturase | *mat*K | 38.60–38.80 | 32.00 | 27.60–27.80 | 32.73–32.87 | 46.71–47.22 | HD03 (38.60), others (38.80); HD02, HD06, HD08 (27.80), others (27.60) |
| | Envelop membrane protein | *cem*A | 38.36 | 26.72 | 31.47 | 32.18 | 49.65 | |

**Table 2.** *Cont.*

| Gene Category | Gene Group | Gene | GC1 | GC2 | GC3 | GCall | ENC | Plants |
|---|---|---|---|---|---|---|---|---|
| Proteins of unknown function | Hypothetical chloroplast reading frames | *ycf1* | 34.94–35.17 | 28.98–29.14 | 24.76–25.05 | 29.58–29.70 | 45.95–46.33 | HD01, HD03, HD09, HD10 (35.06), HD02, HD06, HD08 (35.01), HD04 (35.17), HD05 (34.94), HD07 (35.11); HD01, HD06, HD08~HD10 (29.14), HD02, HD04 (28.98), HD03 (29.08), HD05 (29.06), HD07 (28.92);HD01, HD08~HD10 (24.81), HD02, HD06 (24.76), HD03 (24.92), HD04 (24.97), HD05 (25.05), HD07 (24.87) |
| | | *ycf2* | 41.60–41.63 | 34.34–34.38 | 37.09–37.11 | 37.69–37.70 | 53.31–53.35 | HD03, HD04, HD06 (41.60), HD01, HD02, HD05, HD07~HD10 (41.63);HD01, HD05, HD07~HD10 (34.38, 37.09), HD02 (34.34, 37.09), HD03, HD04, HD06 (34.37, 37.11) |
| | | *ycf3* | 47.93 | 38.46 | 28.99 | 38.46 | 56.67 | |
| | | *ycf4* | 43.78 | 41.08 | 28.65–29.19 | 37.84–38.02 | 46.61–46.79 | HD01, HD07, HD09, HD10 (28.65), others (29.19) |
| | average | | 45.72–45.76 | 37.98–38.00 | 28.54–28.59 | 37.42–37.44 | 48.48–48.51 | |

The CDSs of all samples included 27 photosynthesis genes, 18 self-replication genes, 4 other genes, and 4 genes encoding proteins of unknown function; among them, the gene *accD* encoded the subunit of acetyl-CoA-carboxylase, the key enzyme in fatty acid synthesis. Examination of all samples separately revealed that the CDSs differed in the GC1, GC2, GC3 and GCall contents and ENC values. Examining the same CDS among different samples showed that 22 genes (*atpA, atpE, atpI, cenA, clpP, ndhC, petA, petB, petD, psaB, psbC, psbD, rbcL, rpl14, rpl2, rpl22, rps12, rps14, rps3, rps4, rps7,* and *ycf3*) had consistent GC contents among codons, representing 41.51% of all CDSs. Seven genes (*accD, ndhF, ndhK, rpoB, rpoC2, ycf1* and *ycf2*) showed differences at each codon position, representing 13.21% of all CDSs. Two genes (*ndhG* and *rps8*) showed a difference only in the content of GC1. Three genes (*ndhE, ndhI* and *rpl16*) showed a difference only in the content of GC2. Thirteen genes (*atpB, atpF, ccsA, ndhA, ndhB, ndhD, ndhJ, psaA, psbB, rpl20, rps18, rps2* and *ycf4*) showed a difference only in the content of GC3, accounting for 24.53% of all CDSs. The genes with differences in the contents of GC1 and GC3 included *matK* and *rpoC1*. The genes showing differences in the contents of GC2 and GC3 included *ndhH* and *rpoA*. Therefore, the GC contents varied among the different positions of codons or CDSs, and differences in GC3 among the CDSs were common. Moreover, the expression frequency was different among the CDSs. However, the GC content of the same CDS was the same or only slightly different among all samples, and the average GCall contents of all samples ranged from 37.42 to 37.44. These results indicated that the base composition of codons showed a preference for A or T, especially for the codons whose third base was A or T, which was the vast majority of all codons. Thus, the base composition of all CDSs was highly genetically conserved among all samples.

The ENC values of the different CDSs ranged from 35.23 to 56.67, and the average ENC values of the different samples ranged from 48.48 to 48.51. As the ENC values were over 35, the SCU bias of all samples was weak. Some CDSs showed the same ENC value among different samples, while some samples had the same average ENC values, indicating conservation of gene expression frequency and SCU bias among the samples.

When the differences among all samples were analyzed, several different genes from samples HD03~HD08 exhibited unique variance patterns in the GC content. For example, such a variance was observed for the GC3 contents of the *atpF* and *ndhJ* genes of *C. gigantocarpa* (HD03), the GC2 content of the *rpl16* gene, the GC3 contents of the *rps11* and *rps18* genes of *C. semiserrata* (HD06), the GC2 content of the *ndhE* gene of *C. oleifera* (HD07), and the GC3 contents of the *ccsA, ndhD,* and *rpl20* genes of *C. osmantha* (HD08). The other samples showed no difference in the corresponding GC contents, suggesting species specificity. The samples of *C. gauchowensis, C. vietnamensis* and undetermined species from Hainan province (HD01, HD02, HD09 and HD10) were similar to the sample of *C. oleifera* (HD07) in terms of the GC contents of the different CDSs. In particular, the samples of *C. gauchowensis* from Gaozhou city, *C. vietnamensis* and the undetermined species from Hainan province (HD01, HD09 and HD10) were more similar to each other, and the sample of *C. osmantha* (HD08) was more similar to the samples of *C. gauchowensis, C. vietnamensis* and the undetermined species from Hainan province (HD01, HD02, HD09 and HD10) than to the other samples.

The comparison of the average contents of GC1, GC2 and GC3 among all samples is shown in Figure 1. The GC1, GC2 and GC3 contents of all samples showed almost no differences. Specifically, the GC1, GC2 and GC3 contents were lower and were less than 50%, 40% and 30%, respectively, which indicated that different genes were expressed at different frequencies among all samples, and the base composition of the codons showed a preference for A or T. Finally, genetic conservation was observed among all samples.

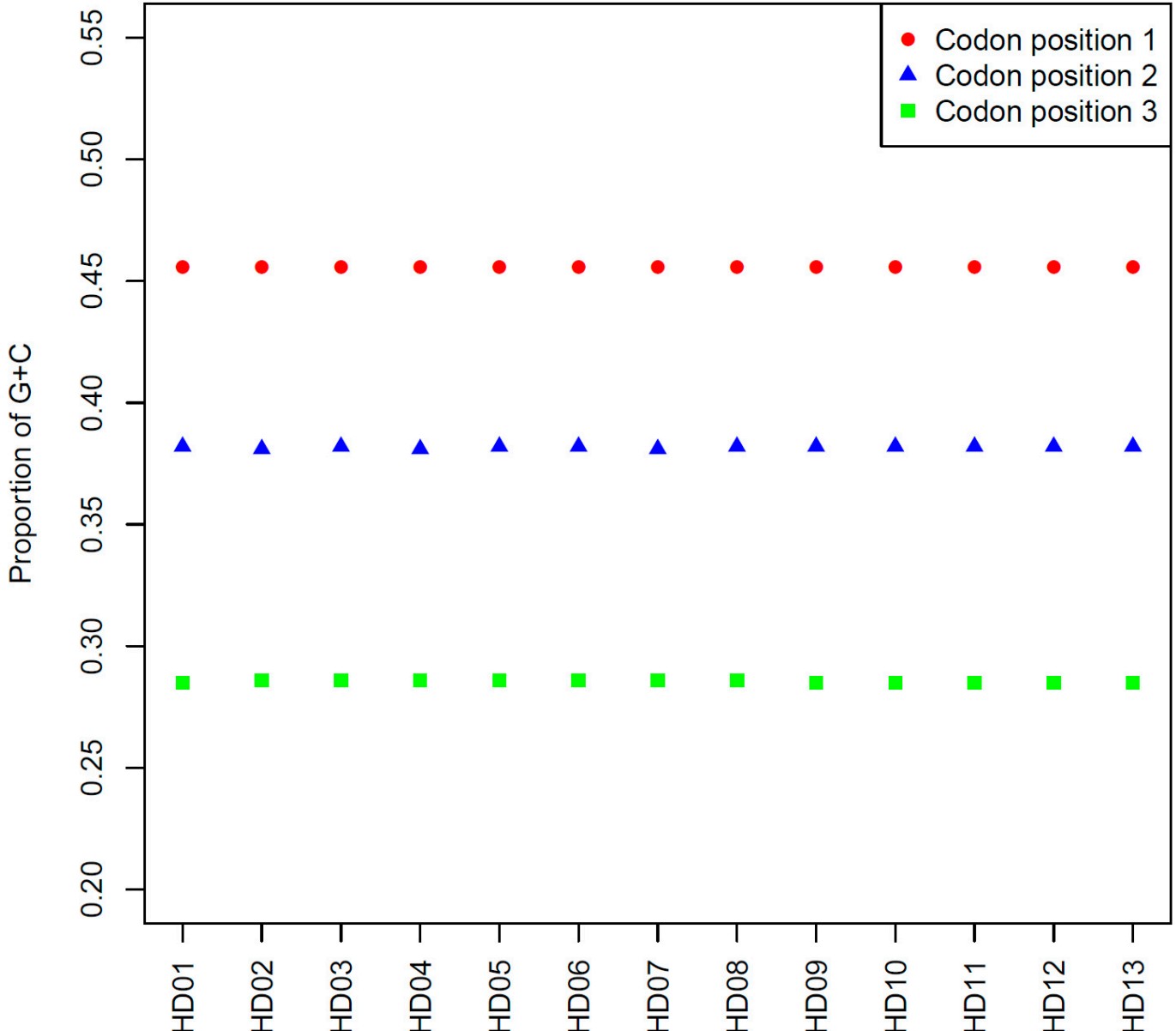

**Figure 1.** Comparison among the average contents of GC at the different codon positions in the cpDNA of the various samples.

The results of one-unit linear correlation analyses between the GC contents of all codon sites, the GCall contents and the ENC values of all CDSs in each sample are shown in Table 3. There were extremely significant linear correlations between GC1 and GC2; GCall and GC1, GC2, and GC3; GCall and GC12; and GC12 and GC1 or GC2. However, the correlations of GC3 with GC1, GC2 and GC12 were not significant. This result indicated that the base compositions at the first and the second positions of codons were similar, yet those at the third position were significantly different from those of the former two. This result was consistent with the previous result of a preference for A or T at the third position. The ENC value was extremely significantly correlated with the content of GC3 but not with the GC1, GC2 and GC12 contents, indicating that the third position of the codons was strongly influenced by SCU bias. Moreover, this finding was consistent with the third position of the codons showing a strong preference for A or T.

**Table 3.** Correlation analysis of the GC content and ENC value of different codon positions in oil-tea camellia.

|  | GC1 (%) | GC2 (%) | GC3 (%) | GCall (%) | GC12 (%) |
|---|---|---|---|---|---|
| GC2 (%) | 0.4430–0.4460 ** | - | - | - | - |
| GC3 (%) | 0.099–0.1230 | −0.0090–−0.0020 | - | - | - |
| GCall (%) | 0.8310–0.8340 ** | 0.7640–0.7670 ** | 0.3900–0.4020 ** | - | - |
| GC12 (%) | 0.8638–0.8667 ** | 0.8326–0.8360 ** | 0.0587–0.0731 | 0.9413–0.9427 ** | - |
| ENC | 0.1420–0.1590 | −0.1720 | 0.3290–0.3420 * | 0.1020–0.1100 | −0.00074–0.0036 |

Note: The symbol ** shows the significance at $p < 0.01$, and the symbol * shows the significance at $p < 0.05$.

### 3.2. Analysis of the RSCU of Oil-Tea Camellia cpDNAs

The RSCU of all cpDNA samples is shown in the stacked bar chart in Figure 2. The stacked bars of all the different samples were highly similar to each other in shape, indicating that all sample cpDNAs were highly genetically conserved. There were 30 high-frequency codons whose RSCU was over 1: the UUU codon of Phe; the Leu synonymous codons UUA, CUU and UUG; the AUU codon of Ile; the Val synonymous codons GUA and GUU; the UAU codon of Tyr; the Gly synonymous codons GGA and GGU; the AAU codon of Asn; the CAA codon of Gln; the AAA codon of Lys; the GAU codon of Asp; the GAA codon of Glu; the Ser synonymous codons UCU, AGU and UCA; the Pro synonymous codons CCU and CCA; the Thr synonymous codons ACU and ACA; the Ala synonymous codons GCU and GCA; the UGU codon of Cys; the CAU codon of His; the Arg synonymous codons AGA, CGA and CGU; and the termination codon UAA. Among the 30 high-frequency codons, U, A and G were the third base in 15, 14 and 1, respectively, indicating that the third position of codons showed a preference for A or U. Otherwise, the codons whose third bases were G or C were all low-frequency codons because their RSCU values were less than 1. Therefore, the third position of the codons showed a preference for A or T.

### 3.3. Neutrality Plot Analysis of Oil-Tea Camellia cpDNAs

The neutrality plots of all samples are shown in Figure 3. The diagrams of the different samples were highly similar to each other, and the vast majority of the points were distributed at the same locations, indicating that the cpDNAs of all samples were highly genetically conserved. The GC3 contents of all 53 CDSs of all samples ranged from 19.12% to 37.11%, and the average GC12 contents of all 53 CDSs of all samples ranged from 32.00% to 55.40%, indicating that the third position of the codons was significantly different from the first two positions. Only the points of the genes *cemA* and *ycf2* were almost distributed along the diagonal, and the point of the gene *atpF* was extremely close to the diagonal. Thus, the SCU bias of these three genes was influenced by mutation pressure. However, the points of the other 50 genes were distributed above and farther away from the diagonal, and the point for the *rps11* gene was the farthest from the diagonal. The SCU bias of all these genes was influenced by selection pressure. The one-unit linear regression and coefficients of determination of the GC12 contents against the GC3 contents ranged from 0.0768 to 0.1002 and from 0.0032 to 0.0054, respectively, which indicated that the linear regression and correlation relationships were not significant, and the maximum and minimum of the regression and correlation coefficients among all samples were observed for *C. semiserrata* (HD06) and *C. meiocarpa* (HD05), respectively. All these results indicated that the third position was significantly different from the first two positions in terms of quantity, while the SCU bias of the vast majority of the codons was influenced by selection pressure.

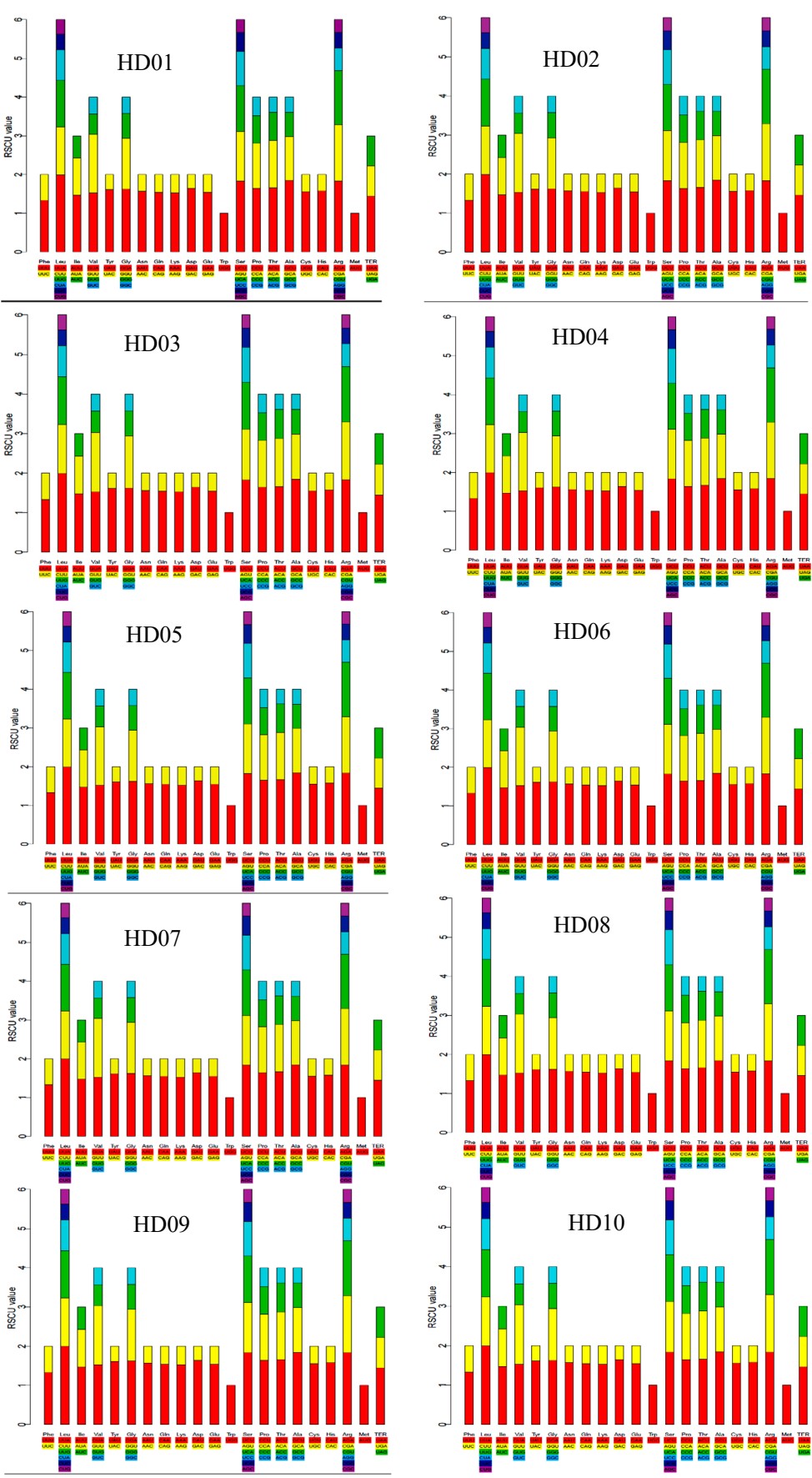

**Figure 2.** RSCU of the codons in the cpDNA of all samples.

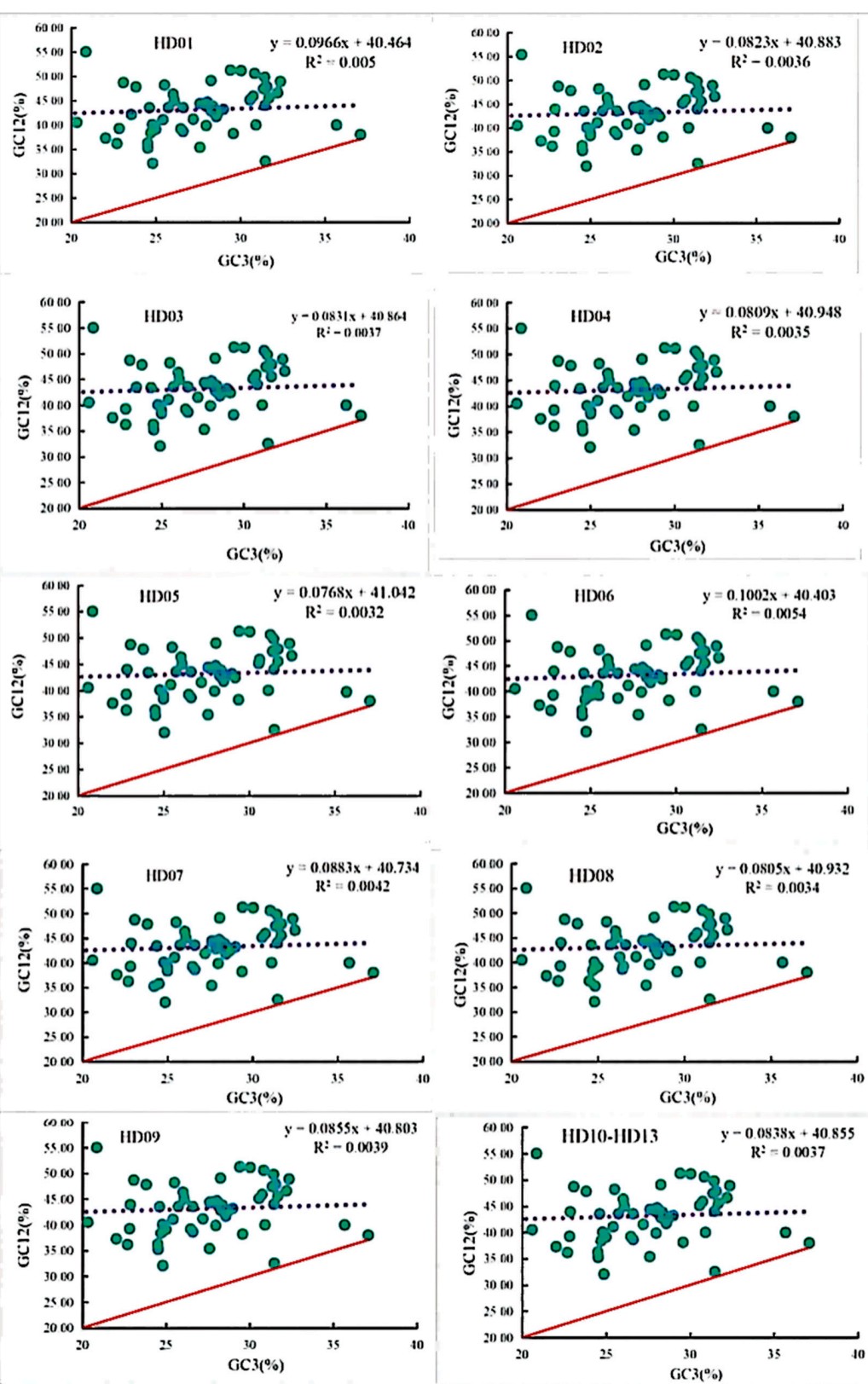

**Figure 3.** Neutrality plot analysis of codons in the chloroplast genomes of oil-tea camellia.

*3.4. ENP Plot Analysis of Oil-Tea Camellia cpDNA*

The results of ENP plot analysis of all cpDNA samples are shown in Figure 4. The diagrams of all samples were highly similar to each other, and the majority of the points were distributed in the lower part of the standard curve, which showed the genetic conservation

of all cpDNA samples. Two genes, *clpP* and *ycf3*, were almost distributed on the standard curve. Additionally, a few genes, including *ndhK*, *rps11*, *rps3* and *rps7*, were extremely close to the standard curve. Therefore, the SCU bias of these 6 genes was mainly influenced by mutation pressure. However, the majority of the points were distributed far away from the standard curve, and the genes *atpF*, *rps14*, *rps18* and *ycf2* were located the farthest from the standard curve, indicating that their SCU bias was mainly influenced by selection pressure. In summary, the SCU bias of oil-tea camellia cpDNAs was mainly influenced by selection pressure, while that of some genes was influenced by mutation pressure.

The ENPexp values and the ratio of ENPexp to ENCcobs of the 53 CDSs of all samples are shown in Table 4, and the ratios showed the same distribution. The frequency of the ratios was clustered in the same frequency distribution chart shown in Table 5, reflecting the genetic conservation of all cpDNA samples. These results were consistent with the results shown in Figure 4. The ratio ranged from −0.051 to 0.051, and the number of genes displayed in Table 5 was 7, accounting for 13.21% of the 53 CDSs. Furthermore, the ratios of genes such as *clpP*, *ndhC*, *clpP* and *ycf3* were almost 0, indicating that the SCU bias of these genes was mainly influenced by mutation pressure, and that of the 4 genes mentioned above was influenced almost exclusively by mutation pressure. As shown in Table 5, the SCU bias of the other 46 genes was influenced more by selection pressure, with the absolute values of the ENC ratios increasing. The ENC ratios of the *atpF*, *rps14* and *rps18* genes were the highest among all CDSs, indicating that their SCU bias was mainly influenced by selection pressure. In summary, the SCU bias of all 13 samples and their impact factors were generally consistent, which reflected the genetic conservation of *Camellia* plants, and the SCU bias of all samples was mainly influenced by selection pressure. However, the influences of mutation pressure cannot be ignored.

**Table 4.** ENCexp value and ENCratio of the chloroplast genome of oil-tea camellia.

| Gene | ENCexp. | ENCratio | Gene | ENCexp. | ENCratio | Gene | ENCexp. | ENCratio |
|------|---------|----------|------|---------|----------|------|---------|----------|
| *accD* | 51.86–52.00 | 0.14–0.15 | *ndhI* | 54.79 | 0.09 | *rpoA* | 54.88–54.20 | 0.10 |
| *atpA* | 50.68 | 0.15 | *ndhJ* | 56.36–56.79 | 0.09–0.10 | *rpoB* | 54.72–54.82 | 0.12 |
| *atpB* | 53.50–53.67 | 0.16 | *ndhK* | 49.12–50.00 | 0.04–0.06 | *rpoC1* | 55.17 | 0.09 |
| *atpE* | 54.09 | 0.12 | *petA* | 55.45 | 0.12 | *rpoC2* | 55.66–55.74 | 0.12 |
| *atpF* | 59.75–59.97 | 0.26–0.27 | *petB* | 52.26 | 0.18 | *rps11* | 45.73–46.54 | −0.05−−0.04 |
| *atpI* | 51.86 | 0.14 | *petD* | 50.64 | 0.14 | *rps12* | 54.75 | 0.08 |
| *ccsA* | 49.28–49.99 | 0.05 | *psaA* | 55.00–55.09 | 0.11 | *rps14* | 56.71 | 0.34 |
| *cemA* | 56.90 | 0.13 | *psaB* | 54.40–54.50 | 0.12 | *rps18* | 52.12–52.98 | 0.33 |
| *clpP* | 49.85 | 0.02 | *psbA* | 53.82–54.04 | 0.25 | *rps2* | 53.06–53.51 | 0.10–0.11 |
| *matK* | 55.19–55.34 | 0.15 | *psbB* | 54.80 | 0.13 | *rps3* | 50.76 | 0.07 |
| *ndhA* | 45.96–46.21 | 0.11 | *psbC* | 54.37 | 0.20 | *rps4* | 53.18 | 0.10 |
| *ndhB* | 54.28–54.43 | 0.14 | *psbD* | 53.86 | 0.20 | *rps7* | 48.81 | 0.06 |
| *ndhC* | 47.52 | 0.03 | *rbcL* | 55.09 | 0.13 | *rps8* | 53.28 | 0.22 |
| *ndhD* | 50.43 | 0.08 | *rpl14* | 52.76 | 0.16 | *ycf1* | 53.16–53.39 | 0.13–0.14 |
| *ndhE* | 50.22 | 0.18 | *rpl16* | 42.20 | 0.17 | *ycf2* | 60.29–60.31 | 0.12 |
| *ndhF* | 47.99–48.10 | 0.13–0.14 | *rpl2* | 57.36 | 0.06 | *ycf3* | 56.27 | −0.01 |
| *ndhG* | 48.54 | 0.12 | *rpl20* | 53.45–54.30 | 0.09 | *ycf4* | 54.19–54.73 | 0.14–0.15 |
| *ndhH* | 48.75–49.26 | 0.05 | *rpl22* | 49.71 | 0.14 | | | |

**Table 5.** Frequency distribution of the ENCratio of the chloroplast genome of oil-tea camellia.

| Class | Lower Limit | Upper Limit | Frequency | Probability (%) | Genes |
|-------|-------------|-------------|-----------|-----------------|-------|
| 1 | −0.051 | 0.051 | 7 | 13.21 | *ccsA, clpP, ndhC, ndhH, ndhK, rps11, ycf3* |
| 2 | 0.051 | 0.153 | 34 | 64.15 | *accD, atpA, atpE, atpI, cemA, rps2, matK, ndhA, ndhB, ndhD, ndhF, ndhG, ndhI, ndhJ, petA, petD, psaA, psaB, psbB, rbcL, rpl2, rpl20, rpoA, rpoB, rpoC1, rpoC2, rps12, rps3, rps4, rps7, ycf1, ycf2, ycf4, rpl22* |
| 3 | 0.153 | 0.255 | 9 | 16.98 | *atpB, ndhE, petB, psbA, psbC, psbD, rpl14, rpl16, rps8* |
| 4 | 0.255 | 0.357 | 3 | 5.66 | *atpF, rps14, rps18* |
| | Total | | 53 | 100.00 | |

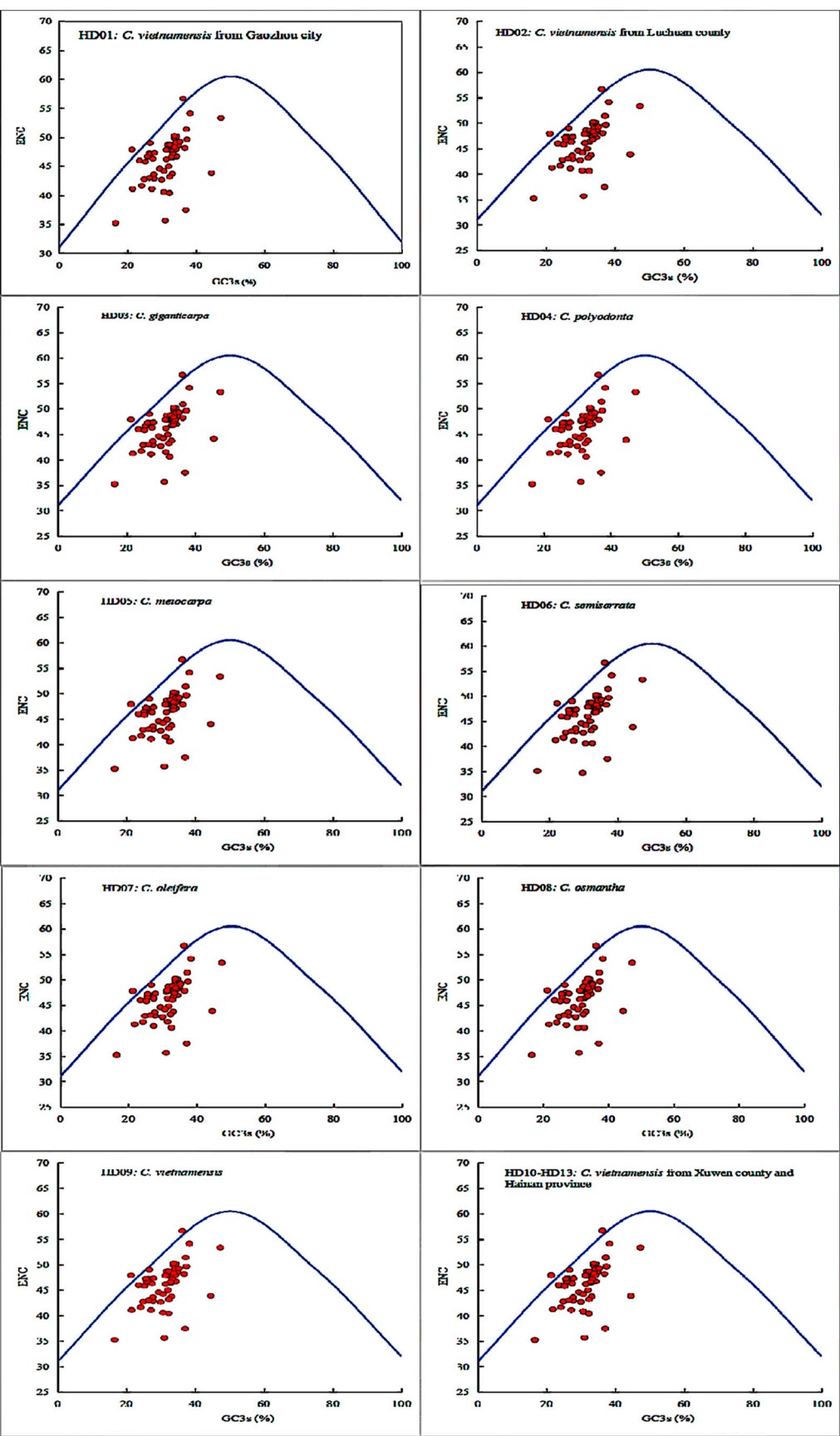

**Figure 4.** ENC plot analysis of codons in the chloroplast genomes of oil-tea camellia.

### 3.5. PR2 Plot Analysis of Oil-Tea Camellia cpDNA

The results of the PR2 plot analysis of all sample codons are shown in Figure 5. The diagrams of all samples were highly similar to each other. The minority of the points drifted slightly, and the third codon position of the genes *ndhH*, *ndhK*, *rpl2* and *rpl14* showed nearly equal use of all 4 bases, while that of the genes *atpF*, *ccsA*, *ndhA*, *ndhB*, *ndhD*, *rpl20*, *rpoA*, *rpoB*, *rpoC1*, *rpoC2*, *rps4*, *rps11*, *rps12* and *ycf3* used A and T evenly and that of the genes *atpA*, *atpB*, *petB*, *petD*, *psbB*, *rpl14*, *ycf1* and *ycf4* used G and C evenly. That of the other genes showed uneven use of the 4 bases, reflecting the genetic conservation of cpDNAs for all samples. The points of all 53 CDSs were distributed in the lower right, lower left, upper left and upper right in descending order, which indicated that the third position used A less than T and C less than G, and the difference between A and T was larger than that between G and C. In general, the SCU bias of all samples was mainly influenced by selection pressure, while the influences of mutation pressure could not be ignored.

### 3.6. Analysis of Optimal Codons in Oil-Tea Camellia cpDNAs

High-expression codons were screened in all samples, and some codons from high- and low-expression gene groups in the cpDNAs of the samples of *C. gauchowensis*, *C. vietnamensis* and the undetermined species from Hainan province (HD01, HD02 and HD09~HD13) showed the same ΔRSCU. A few of the other samples showed less different ΔRSCU values. The details of the results are shown in Table 6.

**Table 6.** Preponderant codon analysis of the chloroplast genome of oil-tea camellia.

| AA | Codon | High_RSCU | Low_RSCU | ΔRSCU | AA | Codon | High_RSCU | Low_RSCU | ΔRSCU |
|---|---|---|---|---|---|---|---|---|---|
| Phe | UUU | 1 | 1.03–1.04 | −0.04−−0.03 | Tyr | UAU | 1.48–1.5 | 1.59 | −0.11−−0.09 |
| | UUC | 1 | 0.96–0.97 | 0.03–0.04 | | UAC * | 0.5–0.52 | 0.41 | 0.09–0.11 |
| Leu | UUA * | 1.59 | 1.21 | 0.38 | TER | UAA * | 2.4 | 0.6 | 1.8 |
| | UUG * | 1.59 | 1.45–1.47 | 0.12–0.14 | | UAG | 0.6 | 1.8 | −1.2 |
| | CUU * | 1.59 | 1.41 | 0.18 | His | CAU | 1.47 | 1.57 | −0.1 |
| | CUC | 0 | 0.54–0.55 | −0.55−−0.54 | | CAC * | 0.53 | 0.43 | 0.1 |
| | CUA * | 1.15 | 0.91 | 0.24 | Gln | CAA * | 1.73 | 1.37 | 0.36 |
| | CUG | 0.09 | 0.46 | −0.37 | | CAG | 0.27 | 0.63 | −0.36 |
| Ile | AUU * | 1.32 | 1.24 | 0.08 | Asn | AAU | 1.1 | 1.51 | −0.41 |
| | AUC | 0.84 | 0.76–0.77 | 0.07–0.08 | | AAC * | 0.9 | 0.49 | 0.41 |
| | AUA | 0.84 | 0.99–1.00 | −0.16−−0.15 | Lys | AAA * | 1.73–1.78 | 1.27–1.28 | 0.45–0.50 |
| Met | AUG | 1 | 1 | 0 | | AAG | 0.22–0.27 | 0.72–0.73 | −0.50−−0.45 |
| Val | GUU * | 1.95–2 | 1.22 | 0.73–0.78 | Asp | GAU | 1.11–1.16 | 1.65 | −0.54−−0.49 |
| | GUC | 0 | 0.73 | −0.73 | | GAC * | 0.84–0.89 | 0.35 | 0.49–0.54 |
| | GUA * | 1.85–1.89 | 1.19 | 0.65–0.7 | Glu | GAA * | 1.52–1.57 | 1.30–1.31 | 0.21–0.23 |
| | GUG | 0.11–0.21 | 0.86 | −0.75−−0.65 | | GAG | 0.47–0.48 | 0.69–0.70 | −0.23−−0.21 |
| Ser | UCU * | 1.97 | 1.64 | 0.33 | Cys | UGU * | 2 | 1.35 | 0.65 |
| | UCC | 0.99 | 1.22 | −0.23 | | UGC | 0 | 0.65 | −0.65 |
| | UCA | 0.63 | 1.14 | −0.51 | TER | UGA | 0 | 0.6 | −0.6 |
| | UCG | 0.45 | 0.69 | −0.24 | Trp | UGG | 1 | 1 | 0 |
| Pro | CCU * | 1.76 | 1.32 | 0.44 | Arg | CGU * | 1.41 | 0.98 | 0.43 |
| | CCC | 0.47 | 0.83–0.86 | −0.39−−0.36 | | CGC | 0.21 | 0.34 | −0.13 |
| | CCA | 1.06 | 1.14 | −0.08 | | CGA * | 1.84 | 1.25–1.26 | 0.58–0.59 |
| | CCG | 0.71 | 0.68 | 0.03 | | CGG | 0.21 | 0.57 | −0.36 |
| Thr | ACU * | 1.95 | 1.3 | 0.65 | Ser | AGU * | 1.52 | 1.05 | 0.47 |
| | ACC * | 1.07 | 0.79 | 0.28 | | AGC * | 0.45 | 0.25 | 0.2 |
| | ACA | 0.98 | 1.21 | −0.23 | Arg | AGA | 1.62 | 1.95–1.96 | −0.34−−0.33 |
| | ACG | 0 | 0.7 | −0.7 | | AGG | 0.71 | 0.89–0.91 | −0.20−−0.14 |
| Ala | GCU * | 2.51 | 1.69 | 0.82 | Gly | GGU * | 1.91 | 0.99 | 0.92 |
| | GCC | 0.2 | 0.98 | −0.78 | | GGC * | 0.55 | 0.33 | 0.22 |
| | GCA * | 1.22 | 0.9 | 0.32 | | GGA | 1.17 | 1.71 | −0.54 |
| | GCG | 0.07 | 0.43 | −0.36 | | GGG | 0.37 | 0.97 | −0.6 |

Note: * indicates ΔRSCU ≥ 0.08.

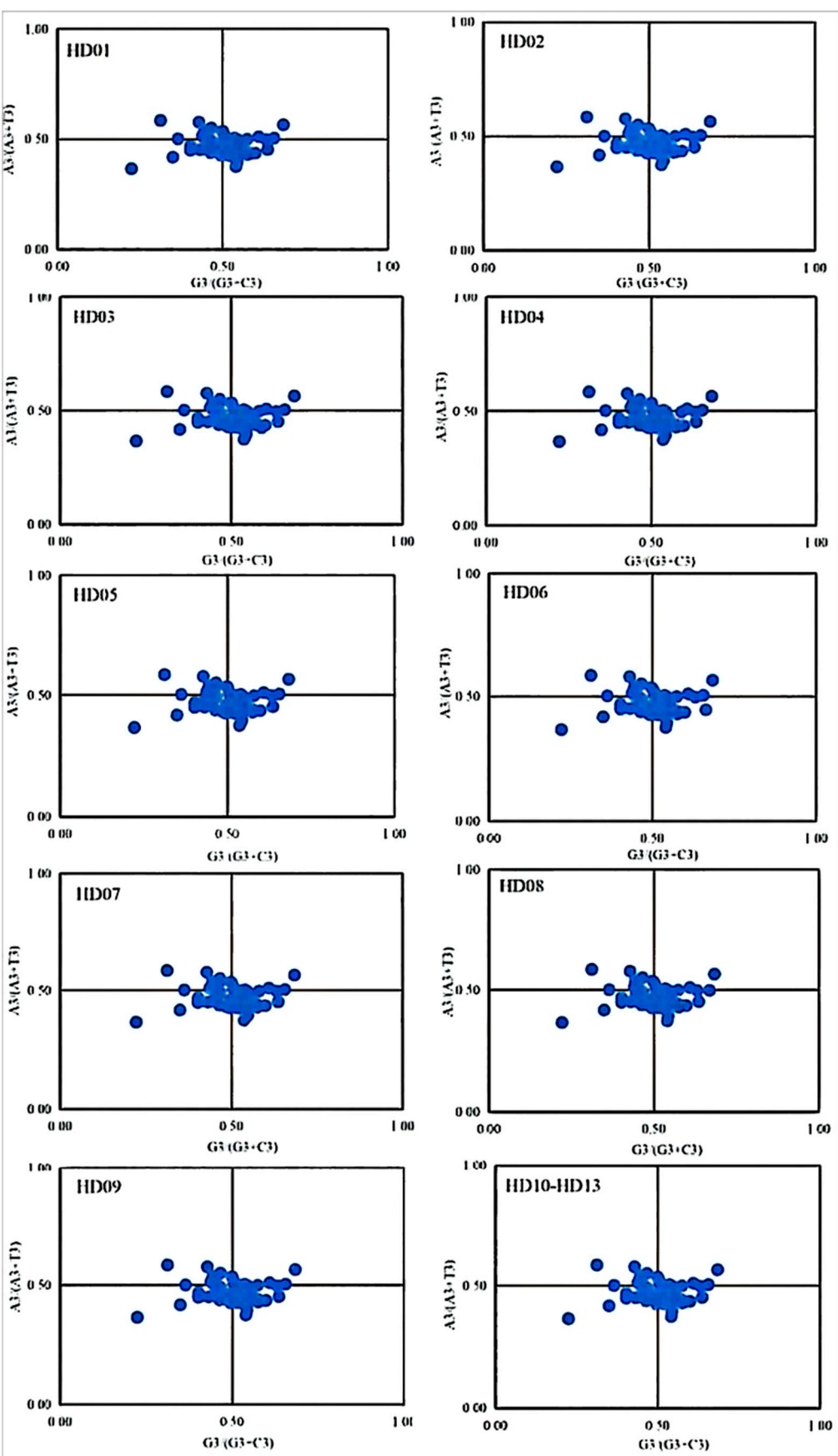

**Figure 5.** PR2-plot analysis of codons in the cpDNA of oil-tea camellia.

The codons with a ΔRSCU ≥ 0.08 could be regarded as high-expression codons. The 28 high-expression codons with the symbol * in Table 6 were screened, and U, A, C and G were the third bases in 11, 9, 7 and 1, respectively. Therefore, the third position of the cpDNA codons showed a preference for A or T. Meanwhile, the SCU bias of cpDNAs appeared to be genetically conserved among the different species of *Camellia*.

The codons found in both the high-expression and high-frequency groups mentioned above included 20 optimal codons. The optimal codons were as follows: codons of Ala including GCU and GCA, codons of Arg including CGA and CGU, the codon of Cys UGU, the codon of Glu GAA, the codon of Gly GGU, the codon of Ile AUU, codons of Leu including UUA, CUU, and UUG, the codon of Lys AAA, the codon of Pro CCU, codons of Ser including UCU and AGU, the codon of Thr ACU, codons of Val including GUA and GUU, and the termination codon UAA. The third base was U, A, C and G in 11, 8, 0 and 1 of these optimal codons, respectively, yet the codons whose third bases were C were not optimal, which indicated that the third position of the optimal codons in the cpDNAs of all samples showed a strong preference for A or T.

### 3.7. Phylogenetic Analysis

A phylogenetic tree was constructed with 10 samples based on the RSCU values and the results are shown in Figure 6. Ten samples clustered into 6 clades ($r^2$ = 0.9196, d = 0.5395). The relationship between *C. vietnamensis* (HD09) and *C. gauchowensis* from Gaozhou city (HD01) was closer than that between *C. gauchowensis* from Gaozhou city (HD01) and *C. gauchowensis* from Luchuan county or Xuwen county (HD02 or HD10), and the relationships between *C. osmantha* (HD08) and *C. gauchowensis* from Gaozhou city (HD01) or Xuwen county (HD10), *C. vietnamensis* (HD09) and the undetermined species from Hainan province (HD10) were closer than that between *C. osmantha* (HD08) and *C. gauchowensis* from Luchuan (HD02). Therefore, these 5 samples were grouped into the same clade. The other samples were separated into independent clades. The relationship with *C. oleifera* (HD07) was progressively weaker for *C. polyodonta* (HD04), *C. meiocarpa* (HD05), *C. semiserrata* (HD06) and other taxa, such as *C. gauchowensis*, *C. vietnamensis*, *C. osmantha* and undetermined species from Hainan province (HD01, HD02, HD08, HD09 and HD10), and the relationship with *C. gigantocarpa* (HD03) was the most distant.

*Hartia laotica* was taken as the outgroup, and cpDNAs of all samples and the other 7 *Camellia* species were used to generate a phylogeny using the Bayesian method (BI) based on the CDSs. The results are shown in Figure 7. The tree divided all samples and the sequences from the NCBI into 2 clades: one included 2 subclades, and the other included 5 subclades. The subclade of *C. granthamiana* included samples of *C. vietnamensis* (HD09), *C. gauchowensis* (HD01, HD02 and HD13), undetermined species from Hainan province (HD10~HD12), *C. osmantha* (HD08) and *C. granthamiana*. The subclade of *C. azalea* included the samples of *C. semiserrata* (HD06) and *C. azalea*. The sample of *C. gigantocarpa* (HD03) formed an independent subclade. The subclade of *C. japonica* or *C. chekiangoleosa* included the samples of *C. polyodonta* (HD04), *C. japonica* and *C. chekiangoleosa*. The subclade of *C. oleifera* included the samples of *C. oleifera* (HD07) and *C. japonica* (the serial number in the NCBI database differed from the one in the above subclade). The subclade of *C. sasanqua* included the samples of *C. sasanqua* and *C. meiocarpa* (HD05). *C. crapnelliana* was an independent subclade.

Figures 6 and 7 show similar relationships among the 13 samples. Notably, for the samples of *C. gauchowensis* and *C. vietnamensis*, the cluster nodes of the population of *C. gauchowensis* from Gaozhou (HD01) and *C. vietnamensis* (HD09) were located at the outermost position, followed by the cluster nodes of these two and the samples of the undetermined species from Hainan province (HD10~HD12) or the population of *C. gauchowensis* from Xuwen (HD13). Finally, the next node included the samples of the population of *C. gauchowensis* from Luchuan (HD02) and *C. osmantha* (HD08); thus, the phylogenetic relationship among *C. vietnamensis*, *C. gauchowensis* and the undetermined species from Hainan province was closer than that between the populations of *C. gauchowensis* from

Gaozhou and Luchuan, meaning that *C. vietnamensis*, *C. gauchowensis* and the undetermined species from Hainan province could be merged into the same species, with *C. osmantha* much closer to them. In summary, because SCU bias varied among the species, this metric can be used to identify the plant species and infer their genetic relationships.

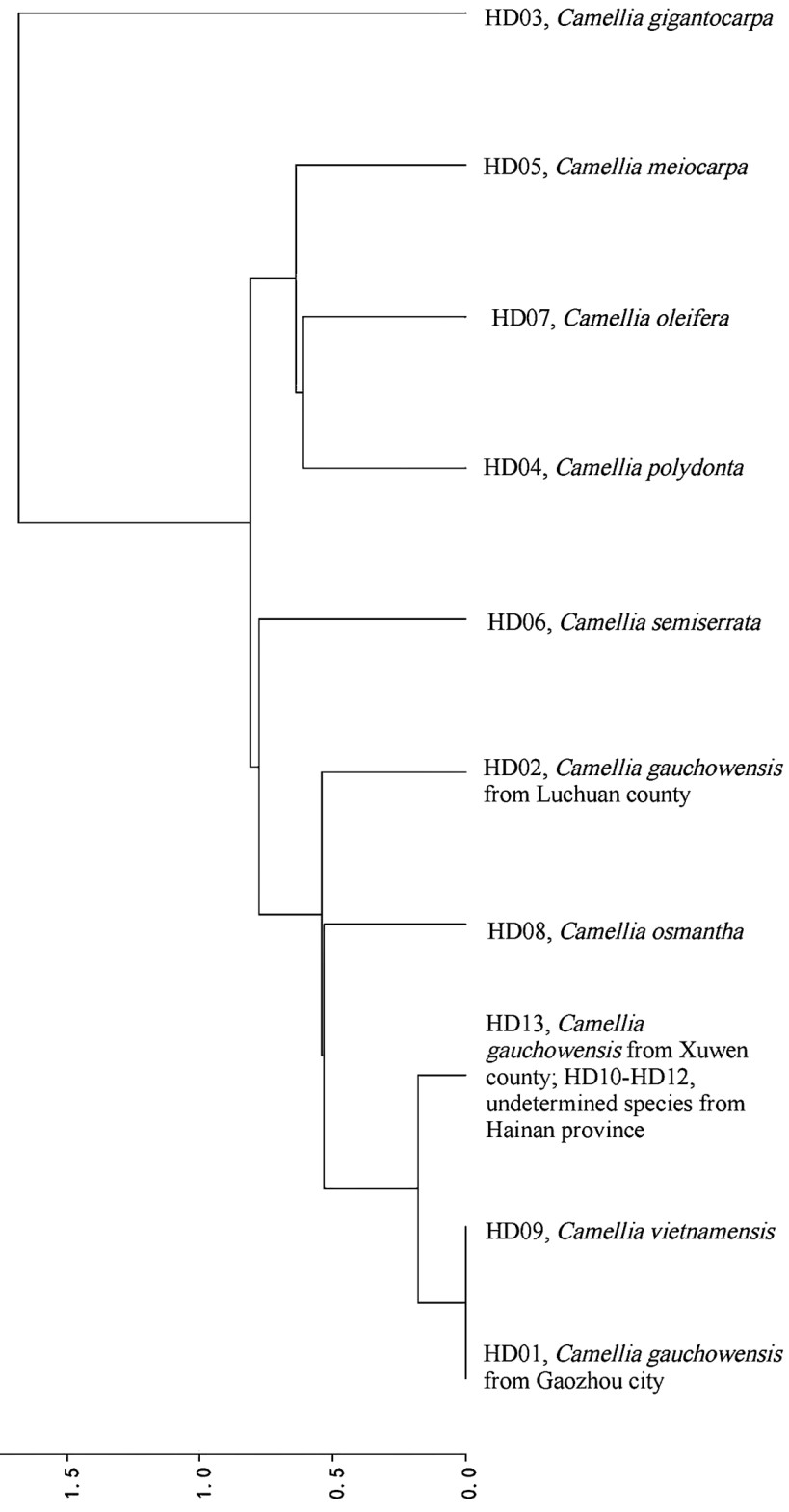

**Figure 6.** Cluster analysis based on the RSCU of the different samples.

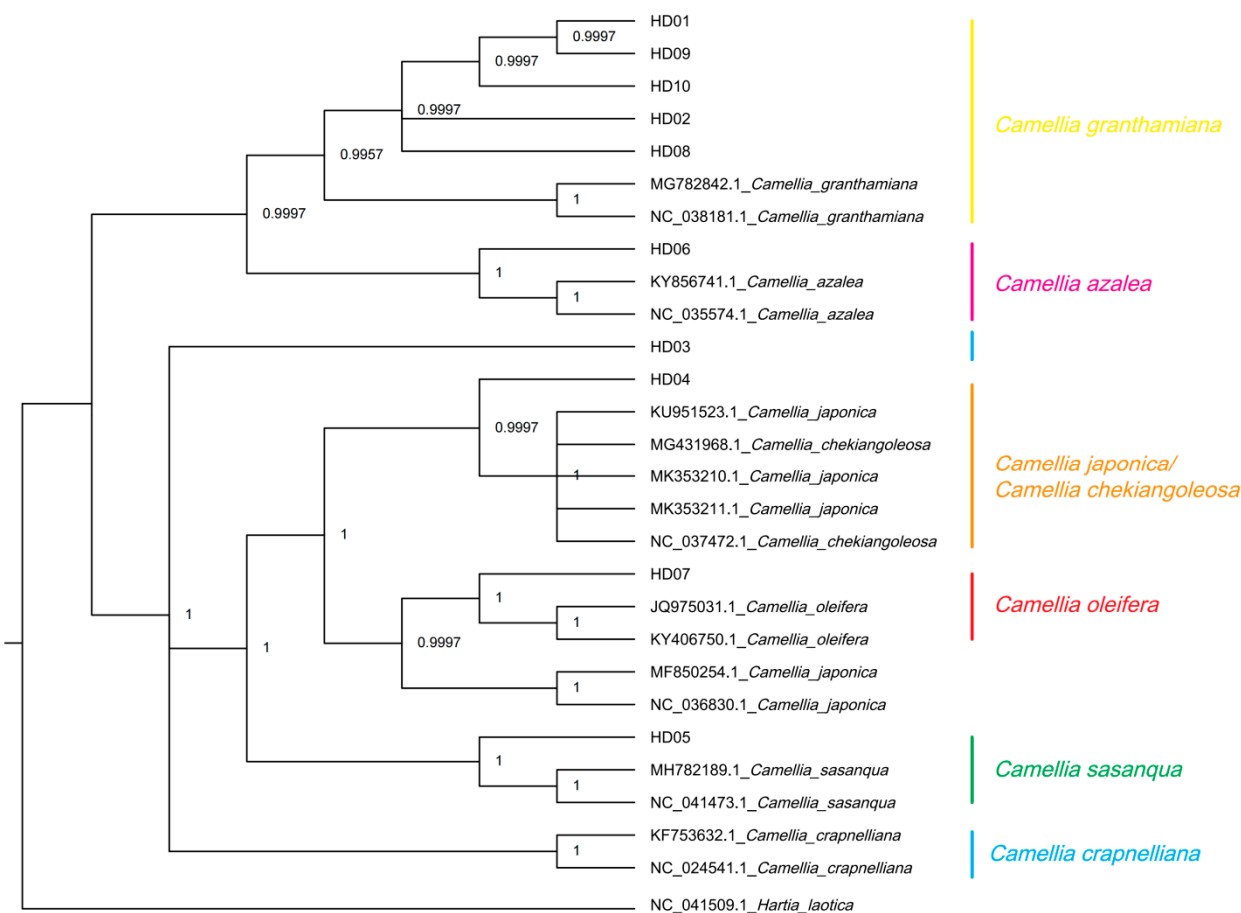

**Figure 7.** Phylogenetic tree of 13 samples based on CDS using the MrBayes method (BI).

## 4. Discussion

### 4.1. The Important Findings of This Paper

In this study, the third position of the codons of oil-tea camellia cpDNAs differed from the first two positions, and they showed a preference for base A or T. This result drove the detected SCU bias, which was influenced by the integrated influences of selection and mutation pressures, especially selection pressure. The third position of high-frequency codons showed an incomplete preference for A or U and occasionally showed a G or C. Similar results were reported in different plants in previous studies, indicating that the cpDNAs were highly genetically conserved and followed similar rules [30–32].

In this study, different species of *Camellia* were used, but the data for the species were highly similar, and the characteristics of SCU bias were consistent. The genes whose high-frequency codons, high-expression codons, optimal codons and SCU bias were mainly influenced by selection pressure were also consistent with each other, indicating that the SCU bias of the different species of *Camellia* was highly genetically conserved. This finding was consistent with a previous report that the cpDNAs of different oil-tea camellia species exhibited good collinearity [19].

Our research group reported that the exons of oil-tea camellia cpDNAs contained some phylogenetic divergence hotspots [19], and the variance in these CDSs was due to SNP site mutations. Meanwhile, SNP mutations at the third position were often synonymous, leading to SCU bias [33,34]. Thus, SCU bias may be used for the evolutionary analysis of species. In fact, the results of evolutionary analysis based on SCU bias and CDSs were consistent with one another, and two similar phylogenetic trees were constructed with 10 samples based on RSCU values and CDSs in this study. These results were consistent with a phylogenetic tree based on the full cpDNAs whose results were related to the samples

described in this paper [19]. The relationships between *C. gauchowensis*, *C. vietnamensis*, the undetermined species from Hainan province and *C. osmantha* were consistent with one another and suggested the merging of *C. vietnamensis* and *C. gauchowensis* into one species. The undetermined species from Hainan province were *C. vietnamensis*, but determining whether *C. osmantha* is an independent species requires further genetic evidence. That *C. gigantocarpa* was separated into an independent clade was also consistent. Therefore, all results related to SCU bias, CDSs and full cpDNAs could be used to distinguish the species and could reveal the genetic relationships among the different species of oil-tea camellia.

### 4.2. Comparison with Previous Similar Reports

Previous studies revealed that *C. oleifera* cpDNA had 18 optimal codons [12], which was inconsistent with the findings presented in this paper. Moreover, because 17 optimal codons were consistent with each other, 3 optimal codons (UUA, CGA and UCU) were screened, and the optimal codon GAC was not observed. The reasons for the differences may be a discrepancy in the defined high- and low-expression gene groups or errors in cpDNA sequencing. This problem should be addressed in further studies to facilitate research on cpDNA gene expression [35–37] and the development of cpDNA genetic engineering [7,36,38].

In this study, the association between SCU bias and specific genes was determined through detailed analysis of a large quantity of data, and then the codon composition and SCU bias of the specific genes were confirmed, distinguishing our study from previous studies that generally ignored specific genes [12,39–41]. The patterns of some specific genes' codon compositions were revealed, and the determinants of the SCU bias of some specific genes were confirmed.

### 4.3. The Value of the SCU Analysis in This Paper

The SCU bias of some specific genes was revealed in this paper, which will support further studies on the regulation of oil-tea camellia cpDNA gene expression [35,37] and the development of oil-tea camellia cpDNA genetic engineering [7,38].

SCU bias could be used to distinguish the species and could reveal the genetic relationships among the different species of oil-tea camellia in this study, indicating that SCU bias reflects species specificity [17,42,43] and could be used to construct a technological system for identifying oil-tea camellia germplasm species.

### 5. Conclusions

Even base composition was observed for very few genes in oil-tea camellia cpDNAs; instead, the codons of the vast majority of the genes showed a preference for A or T, the third position showed a strong preference for A or T, and the base at the third position determined SCU bias. The SCU bias of oil-tea camellia is weak, and it is influenced by selection pressure; meanwhile, the influences of mutation pressure cannot be ignored because 37.74% of all codons showed SCU bias influenced by mutation pressure. Among the oil-tea camellia cpDNAs, 30 high-frequency codons, 28 high-expression codons and 20 optimal codons were screened, and the third base of the codons showed a significant or strong preference for A or U. The characteristics of the codon composition and SCU bias of oil-tea camellia cpDNAs were confirmed, which can support studies of the regulation of oil-tea camellia cpDNA gene expression and the development of oil-tea cpDNA genetic engineering. As the SCU bias of oil-tea camellia cpDNAs is strongly genetically conserved but shows species specificity, the RSCU values of cpDNAs can be used for species identification of oil-tea camellia germplasm. The results suggest merging *C. vietnamensis* and *C. gauchowensis* into one species and that the undetermined species from Hainan province is *C. vietnamensis*. However, determining whether *C. osmantha* is an independent species requires further genetic evidence.

**Author Contributions:** Conceptualization, X.H. and K.Z.; methodology, K.Z. and W.M.; formal analysis, J.C.; investigation, J.C.; resources, J.C.; data curation, J.C. and W.M.; writing—original draft preparation, J.C.; writing—review and editing, X.H. and K.Z.; supervision, X.H. and K.Z.; project

administration, X.H.; funding acquisition, X.H. All authors have read and agreed to the published version of the manuscript.

**Funding:** This research was funded by the Key R&D Program of Hainan Province, China (ZDYF2022SHFZ020).

**Institutional Review Board Statement:** Not applicable.

**Informed Consent Statement:** Not applicable.

**Data Availability Statement:** The data presented in this study are available on request from the corresponding author. The data are not publicly available due to privacy.

**Conflicts of Interest:** The authors declare no conflict of interest.

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
