# Peer review of "Synonymous Codon Usage Bias in the Chloroplast Genomes of 13 Oil-Tea Camellia Samples from South China"

_forests, doi:10.3390/f14040794_

Round 1

Reviewer 1 Report

1.       The introduction is brief; explain why you want to conduct this study, any previous studies in other plant species, including Camellia, why you chose Camellia, what the evolutionary significance of 13 selected Camellia plant samples is, and so on.

Results:

1.       Although you have mentioned it in the text, it would be helpful if you could mark the category of genes in the table so that the reader can understand what types of genes you have studied.

2.       The 53 efficient coding sequences (CDSs) were mostly photosynthesis genes (27), self-replication genes (18) and 4 with unknown function. Is there any evidence to get gene related to oil biosynthesis pathways? If not, then what could be the reason.

3.       Table 2, is there any way to use statistics to compare the GC1/GC2/GC3 across the different samples?

Author Response

Respected reviewer,      

       Greetings!  We have modified our manuscript according to your guidance, and we had sent the manuscript to the English native speakers to be polished for us. Please check the modified manuscript!  thank you for your guidance again!  Our response to your comments are as follows:

  1. The introduction is brief; explain why you want to conduct this study, any previous studies in other plant species, including Camellia, why you chose Camellia, what the evolutionary significance of 13 selected Camellia plant samples is, and so on.

          Answer: Thank you very much for your guidance! We have tried to add                large quantities of contents in the section of introduction according to                your guidance.

         Results:

  1. Although you have mentioned it in the text, it would be helpful if you could mark the category of genes in the table so that the reader can understand what types of genes you have studied.

          Answer: Thank you very much for your guidance! We have added two                  columns of “Category for genes” and “Group of genes” respectively,                      meanwhile, we delete some genes names in the text in order to avoid                  repeat.

  1. The 53 efficient coding sequences (CDSs) were mostly photosynthesis genes (27), self-replication genes (18) and 4 with unknown function. Is there any evidence to get gene related to oil biosynthesis pathways? If not, then what could be the reason.

          Answer: Thank you very much for your guidance! We have found the                    gene   accD who code the subunit of acetyl-CoA-carboxylase, and this                  enzyme is the key enzyme of the synthesis of fatty acid, then we have                  added this content to the text.

  1. Table 2, is there any way to use statistics to compare the GC1/GC2/GC3 across the different samples?

         Answer: We must say sorry to you because we only made descriptive                   statistics analysis! Surely we should design some replications to analysis               the significance of the difference among the different samples, so we                   regretted there were only 13 samples standing for the different species or           the different populations. Thank you very much for your guidance!

Kind regards!

 Kaibing Zhou and other authors

Reviewer 2 Report

The work is undoubtedly interesting, although some parts are difficult to follow. It is written correctly, but it is not fluent, and there are numerous repetitions of expressions. For instance, the term "were consistent" is used repeatedly throughout the text. All scientific names should be italicized, including the genus Camellia and all Camellia species listed (see, for example, lines 41 and 42 of the introduction).

The words in the title (oil-tea camellia; synonymous codon usage bias) should be removed from the keywords and replaced.

Lines 51-54 contain an unclear statement that does not show any connection with environmental effects and cultivation technologies. The authors must clarify their reasoning. The same applies to the statement in line 63, as it is not apparent how it facilitates the formulation of cultivation technology.

"Sample tree's site" in Table 1 is not very clear, and I suggest replacing it with a more descriptive phrase. The same goes for Table 2, where I recommend changing "different samples" to a more precise term.

Line 60: countries.

Author Response

Respected reviewer,    

         Greetings!

         We have modified our manuscript according to your guidance, and we had sent the manuscript to the English native speakers to be polished for us. Please check the modified manuscript!  Thank you for your guidance again! Response to your comments are as follows:

          The work is undoubtedly interesting, although some parts are difficult to follow. It is written correctly, but it is not fluent, and there are numerous repetitions of expressions. For instance, the term "were consistent" is used repeatedly throughout the text.

          Answer: Thank you very much for your guidance! We had sent this paper to an English native speaker to polish.

          All scientific names should be italicized, including the genus Camellia and all Camellia species listed (see, for example, lines 41 and 42 of the introduction).

          Answer: Thank you very much for your guidance! We have made all scientific names italicized, additionally the genes names too.

          The words in the title (oil-tea camellia; synonymous codon usage bias) should be removed from the keywords and replaced. 

           Answer: Thank you very much for your guidance ! We have replaced the keywords “oil-tea camellia” and “synonymous codon usage bias” into Camellia” and “RSCU” respectively.

            Lines 51-54 contain an unclear statement that does not show any connection with environmental effects and cultivation technologies. The authors must clarify their reasoning. The same applies to the statement in line 63, as it is not apparent how it facilitates the formulation of cultivation technology.

            Answer: We must say sorry to you because we mistook the spirits of the reference, so we have deleted the sentences from line 51 to line 54 and from line 63 to line 64. Thank you very much for your guidance!

           "Sample tree's site" in Table 1 is not very clear, and I suggest replacing it with a more descriptive phrase. The same goes for Table 2, where I recommend changing "different samples" to a more precise term.

             Answer: Thank you very much for your guidance! We changed both of “sample tree” and “different samples” into “plant”.

             Line 60: countries.

             Answer: Thank you very much for your guidance! We have changed the sentence into -whose samples were collected from production areas in 7 different counties or cities. Namely, we added the words “different” and “or cities”.

Kind regards!

 Kaibing Zhou and other authors
